# HARMONIC: Harnessing LLMs for Tabular Data Synthesis and Privacy Protection

**Yuxin Wang**
Sichuan University
Chengdu, China
wangyuxin1st@gmail.com

**Duanyu Feng**
Sichuan University
Chengdu, China
fengduanyuscu@stu.scu.edu.cn

**Yongfu Dai**
Sichuan University
Chengdu, China
wal.daishen@gmail.com

**Zhengyu Chen**
Wuhan University
Wuhan, China
2019302120293@whu.edu.cn

**Jimin Huang**
The Fin AI
Singapore
jimin.huang@thefin.ai

**Sophia Ananiadou**
The University of Manchester
Manchester, UK
sophia.ananiadou@manchester.ac.uk

**Qianqian Xie**[*]
The Fin AI
Singapore
qianqian.xie@thefin.ai

**Hao Wang**[*]
Sichuan University
Chengdu, China
wangh@scu.edu.cn

## Abstract

Data serves as the fundamental basis for advancing deep learning. The tabular data presented in a structured format is highly valuable for modeling and training. However, even in the era of LLM, obtaining tabular data from sensitive domains remains a challenge due to privacy or copyright concerns. Therefore, exploring the methods for effectively using models like LLMs to generate synthetic tabular data, which is privacy-preserving but similar to original one, is urgent. In this paper, we introduce a new framework HARMONIC for tabular data generation and evaluation by LLMs. In the data generation part of our framework, we employ fine-tuning to generate tabular data and enhance privacy rather than continued pre-training which is often used by previous small-scale LLM-based methods. In particular, we construct an instruction fine-tuning dataset based on the idea of the k-nearest neighbors algorithm to inspire LLMs to discover inter-row relationships. By such fine-tuning, LLMs are trained to remember the format and connections of the data rather than the data itself, which reduces the risk of privacy leakage. The experiments find that our tabular data generation achieves equivalent performance as existing methods but with better privacy by the metric of MLE, DCR, etc. In the evaluation part of our framework, we develop a specific privacy risk metric DLT for LLM synthetic data generation, which quantifies the extent to which the generator itself leaks data. We also developed LLE, a performance evaluation metric for downstream LLM tasks, which is more practical and credible than previous metrics. The experiments show that our data generation method outperform the previous methods in the metrics DLT and LLE.

---

[*]Co-Corresponding Author.

# 1   Introduction

In the age of deep learning, tabular data is a predominant data format and a key element for building more effective algorithms to solve specific applications in various fields [1, 2]. However, in many sensitive domains such as business [3], healthcare [4], and governmental operations [5], there are significant limitations on the acquisition and use of tabular data. Tabular data in these domains involves personal privacy, business secrets, or state secrets. The collection and use of such data are strictly regulated by laws and regulations, and compliance with relevant data protection requirements is necessary. Unauthorized use or disclosure may result in serious privacy infringement or business losses. Therefore, generating data that ensures the effectiveness in modeling these data while preserving privacy in tabular data synthesis has always been a critical research area [6, 7, 8].

Traditionally, Tabular data synthesis often rely on methods like GANs [9, 10, 11], VAEs [12, 13], and Diffusion Models [14, 15, 16, 17]. However, the rise of Large Language Models (LLMs) with their impressive ability to generate data has shifted the paradigm. Methods like GReaT [18] and TabuLa [6] leverage LLMs for faster synthesis by converting tables to natural language and fine-tuning the LLMs through next-word prediction to get a generator. They often utilize smaller pre-trained models like GPT-2 [19] for efficiency. Despite their powerful language understanding abilities, LLMs introduce significant privacy concerns [20, 21]. Continued pre-training methods may exacerbate this tendency to leak original data. Therefore, a crucial area of exploration lies in developing strategies to mitigate these privacy risks while harnessing the power of LLMs for tabular data synthesis.

To Harness LLMs fOr Tabular Data SyNthesis and PrIvacy ProteCtion, we develop a new framework, HARMONIC[2], for the generation of tabular data by LLMs and its evaluation. For generation of the tabular data, we use existing larger-scale LLMs to leverage their in-context learning abilities for generating tabular data while ensuring privacy by fine-tuning. To be precise, we employ the idea of k-nearest neighbor algorithm (kNN) [22] to construct the instruction fine-tuning datasets. This allows the LLMs to see the relationship between multiple similar rows and construct the structural tabular synthetic data format. This dataset with this format then retain more structural information for LLMs to enhance the ability to generate synthetic data by fine-tuning but avoid the forced memorization of data with pre-training. For the comprehensive evaluation of the synthetic data generated by LLMs, especially its effectiveness and privacy, we introduce two novel metrics: LLE (LLM Effectiveness) and DLT (Data Leakage Test), where LLE evaluates the effectiveness of the synthetic data in downstream LLM tasks while DLT quantifies the privacy risk by comparing the perplexity of the generator on original data and synthetic data.

We assess our HARMONIC data generation framework together with existing methods of data generation by four datasets commonly used for classification tasks in tabular data synthesis, using both representative metrics and DLT and LLE. The results show that the data generated by HARMONIC performs comparably to existing methods in effectiveness but excels in privacy assessments. Crucially, HARMONIC's evaluation suggests that traditional synthetic data methods may be unsuitable for downstream LLM tasks and that pretraining-based synthetic data may pose greater privacy risks.

The main contributions of this study can be summarized as follows: 1) We recognize that it is crucial to not only focus on the strong data generation ability of LLM in this era, but also pay attention to the potential privacy risks it may bring. 2) We develop a framework, HARMONIC, for synthesizing tabular data based on LLM. The framework aims to minimize the risk of data leakage while ensuring the effectiveness of data synthesis using LLM. 3) Under the HARMONIC framework, a set of metrics is proposed for the effectiveness in downstream LLMs tasks and privacy risk evaluation of synthetic tabular data.

# 2   Related work

**Tabular Data Synthesis**. Prior to the rise of Large Language Models (LLMs), synthetic tabular data generation primarily relied on machine learning or classical neural network frameworks. These methods can be broadly categorized into three groups: Generative Adversarial Networks (GANs), Variational Autoencoder (VAE), and Diffusion Models. Building on VAE, TVAE [9] introduces a conditional generator with variational autoencoder (VAE) to generate tabular data. With the

---

[2]`https://github.com/Wendy619/HARMONIC`.

framework of GANs, CTAB-GAN [10] tackles data imbalance and long-tail issues. For Diffusion-based methods, TabDDPM [14] serves as a prominent benchmark, and TABSYN [15] offering faster synthesis compared with other such techniques. In addition to these three categories, early method like SMOTE [23] can also leverage linear interpolation for data generation. However, most of these methods utilize one-hot encoding for categorical data, which can exacerbate the "curse of dimensionality" for high-cardinality variables and fail to capture contextual information [18, 6]. Additionally, these methods overlook the semantic information present in tables.

LLMs have emerged as a compelling approach for synthetic data generation due to their exceptional capabilities in producing high-effectiveness, human-like data. LLM-based methods commonly employ a continued pre-training paradigm, and the original tabular data is converted into text format and fed into the LLM for learning. GreaT [18] exemplifies this approach, converting each tabular feature into the format "X is Y" and feeding the text into GPT-2 [19] with training. REaLTabFormer [24] separate the table into parent table and child table as another format, and also use GPT-2 with continued pre-training and fine-tuning for synthetic data generation. Tabula [6] futher uses the power of this pre-training and fine-tuning process, and prioritizes faster training speed by simplifying token sequences to "X Y". While LLM-based methods often outperform machine learning approaches due to their ability to leverage contextual information, limitations exist. Processing table data row-by-row hinders LLMs from fully exploiting relational information between samples. Furthermore, inherent security risks associated with data leakage plague LLMs [20]. Pre-training method may make them vulnerable, potentially allowing an attacker with knowledge of one or two feature values in a row of original data to retrieve the entire original data record.

**Tabular Data Synthesis Protection.** To enhance the privacy protection for synthetic methods, most existing approaches incorporate differential privacy modules with the existing synthetic methods. For example, PrivBayes [25], CTAB-GAN+ [26], DP-TBART [27] and Mattern et al. [28] use the differential privacy with Bayes, CTABGAN, Bart and GPT-2 for synthetic data and its leakage protection, respectively. While these differential privacy methods offer a path towards privacy preservation on top of existing data synthesis methods, they often impose strict limitations on data features, leading to a significant decrease in downstream model performance. Therefore, we aim to enhance data leakage protection directly from the data synthesis method itself, minimizing the impact on downstream model effectiveness. Moreover, it's important to note that our method does not conflict with these differential privacy methods and can even be further integrated with them to achieve stronger privacy guarantees.

**Tabular Data Synthesis Evaluation**. Beyond evaluating the statistical distribution characteristics of synthetic data [29], existing evaluation methods for synthetic data, such as the MLE benchmarking system proposed by Xu et al. [9], primarily focus on assessing its performance as training data for machine learning models. However, as Kotelnikov et al. [14] argue, relying on weak classifiers for evaluation becomes outdated in light of the capabilities of advanced models like CatBoost [30]. This underscores the need for more sophisticated evaluation techniques, especially considering the widespread adoption of LLMs in downstream applications [31].

Current privacy metrics for synthetic data, such as Distance to Closest Record (DCR) [10] and the NewRowSynthesis metric from SDMetrics [32], solely analyze the distance between synthetic data and original data. While these distance-based approaches provide valuable insights, they fall short when dealing with Large Language Models (LLMs). LLMs are particularly susceptible to data leakage due to their complex nature and training on massive datasets [20]. However, existing privacy metrics based solely on tabular data feature distances fail to capture the unique learning and inference mechanisms of LLMs, which operate at the semantic and generative probability levels of embeddings. Consequently, these methods lack intuitive indicators of privacy leakage specific to LLMs [33].

# 3   HARMONIC Framework

This section is devoted to present the HARMONIC framework for tabular data synthesis powered by LLMs, encompassing both data generation and data evaluation.

## 3.1 Synthetic Tabular Data Generation

We first present our synthetic tabular data generation approach, which utilizes fine-tuning LLMs for the generation of synthetic tabular data. It includes three key stages: (1) **Instruction dataset construction**: Construct an instruction fine-tuning dataset designed to fine-tune the generator model and a prompt dataset to facilitate data generation. (2) **Instruction tuning based tabular data synthesizer formation**: Fed the instruction fine-tuning dataset into an LLM for fine-tuning, as illustrated in Figure 1; (3) **Sampling for synthetic data generation**: Synthetic tabular data is generated by sampling from the fine-tuned LLM, with the sampling process described in Figure 2.

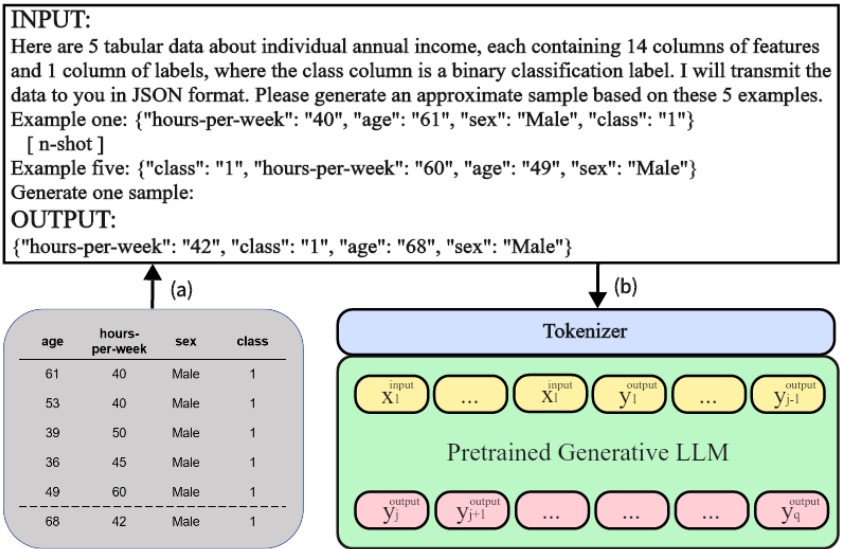

Figure 1: The fine-tuning step. After applying the kNN algorithm to the original table of data, we obtain $n$ sets of $k + 1$ data points. Each set is structured according to the template shown in the gray table at the bottom left. These datasets are then changed to the instructions with the features of each table data shuffled, as shown in the white box above (a). Finally, the fine-tuning dataset is input into the pre-trained LLM for fine-tuning (b).

### 3.1.1 Instruction Dataset Construction

**Construct the instruction fine-tuning dataset using kNN.** Our approach aims to allow LLMs to learn from a few original data instances and generate similar but distinct synthetic data. To achieve this goal, we use the kNN algorithm to identify neighboring data for each instance, enabling LLMs to learn to generate the original data from these neighbors by their in-context learning ability.

Specifically, for each sample in the training set (a row of data in the table), the kNN algorithm is used to find the $k$ nearest neighbors (with a default value of 5) of the sample. This results in $k$ input data points and one label (referred to as a $k + 1$ dataset).

To improve the effectiveness of the generated synthetic data, a filtering step is necessary. For each $k + 1$ dataset, if more than half of the input data have labels that are different from that of the sample, then this $k + 1$ data is discarded. Ultimately, this filtering process yields $n$ sets of $k + 1$ data.

**Data format conversion.** Since LLMs are designed as sequence-to-sequence models, feeding tabular data into an LLM requires converting the structured data into a textual format. A straightforward approach would be to directly input a programming language readable data structure, such as Pandas DataFrame Loader for Python, line-separated JSON-file format, HTML code reflecting tables [1]. In our work, each row of data $s_i$ in a $k + 1$ data set obtained by kNN is converted into JSON dictionary format, preserving the original table structure and enabling the model to understand the semantics of each value.

Specifically, for a row of data $s_i$ in each $k + 1$ data, it has feature names $f_1, f_2, \ldots, f_m$, where the value of its $j$-th feature is $v_{i,j}$. Then, the JSON-formatted data $t_i$ corresponding to $s_i$ is defined as

follows:

$$t_{i,j} = [f_j : v_{i,j}] \qquad \forall i \in \{1, \ldots, n(k+1)\}, j \in \{1, \ldots, m\}, \tag{1}$$

$$\mathbf{t}_i = \{t_{i,1}, t_{i,2}, \ldots, t_{i,m}\} \qquad \forall i \in \{1, \ldots, n(k+1)\}, \tag{2}$$

We concatenate $k$ nearest neighbors JSON-formatted data sequentially, incorporating prompts as the input to elucidate the fine-tuning task. The left row of JSON-formatted data as the reference answer (the output).

In addition, when converting a tabular feature vector, we inadvertently introduce pseudo-positional information into the transformed tabular data. Because there is no inherent spatial ordering among features in tabular datasets [34]. To maintain this independence of the order of the features, we randomly shuffle the order of the features within each row of JSON-formatted data $\mathbf{t}_i$ in the input using a permutation. This operation results in a new sequence where the order of the features is randomized so that the model learns to be invariant to the order of the feature. A template for this instruction fine-tuning dataset is shown as Figure 1. [3]

### 3.1.2 Instruction Tuning Based Tabular Data Synthesizer Formation

We then fine-tune the LLM for the synthetic data generation task using the instruction dataset we constructed. After tokenizing our instruction dataset, the resulting token embeddings of one sample for the input and the output are denoted as $\mathrm{emb}(X) = (x_1, \ldots, x_l)$ and $\mathrm{emb}(Y) = (y_1, \ldots, y_q)$, respectively. Here, $l$ and $q$ represent the lengths of the input and the output, respectively. Therefore, the objective of our fine-tuning strategy is to maximize the probability of generating the correct output sequence given the prompt describing the task and $k$ input original data points. This objective function is formulated as:

$$p(\mathrm{emb}(Y)|\mathrm{emb}(X)) = p(y_1, \ldots, y_q|x_1, \ldots, x_l) = \prod_{j=1}^{q} p(y|x_1, \ldots, x_l, y_1, \ldots, y_j). \tag{3}$$

The LLM is trained by optimizing the parameters to maximize the probability of all the $p(\mathrm{emb}(Y)|\mathrm{emb}(X))$ sample, which only involves minimizing the loss of the output but avoids to learn the original data in the input. We denote the fine-tuned LLM as the generator $\mathbf{G}$ for tabular data synthesis.

### 3.1.3 Sampling for Synthetic Data Generation

To generate the synthetic data by LLMs, we construct a prompt dataset consistent in format with the fine-tuning dataset, where each data point consists of $k$ original data that are randomly resampled from the original data. We emphasize that the data points should be different from those in the fine-tuning dataset which prevent the LLMs from reproducing the original data. Specifically, each data point in the prompt dataset is fed into $\mathbf{G}$, yielding the distribution of subsequent tokens conditioned on the known input sequence. In the end, a full sequence of a synthetic tabular data will be generated. To generate the next token with more diversity and protect privacy, we adopt a weighted sampling strategy that incorporates a temperature coefficient $T$. We set the default temperature coefficient $T$ to 0.7. After generation, we utilize pattern-matching algorithms developed in [35], to reconvert the generated textual feature representations into a dataframe format, resulting in the final synthetic tabular dataset.

### 3.2 Synthetic Tabular Data Evaluation

We introduce two new metrics to evaluate the effectiveness and privacy of synthetic data for LLM-based synthesis methods: LLM Effectiveness (LLE) and Data Leakage Test (DLT).

### 3.2.1 LLE: LLM Effectiveness

With the advancement of LLMs, more and more studies believe that evaluating the effectiveness of synthetic data on the downstream tasks with weak classifiers is losing its practical value and credibility [14]. Recent research find that the application of LLMs to tabular data processing yields

---

[3] For more detail about the format and the prompt, please refer to Appendix A.5.

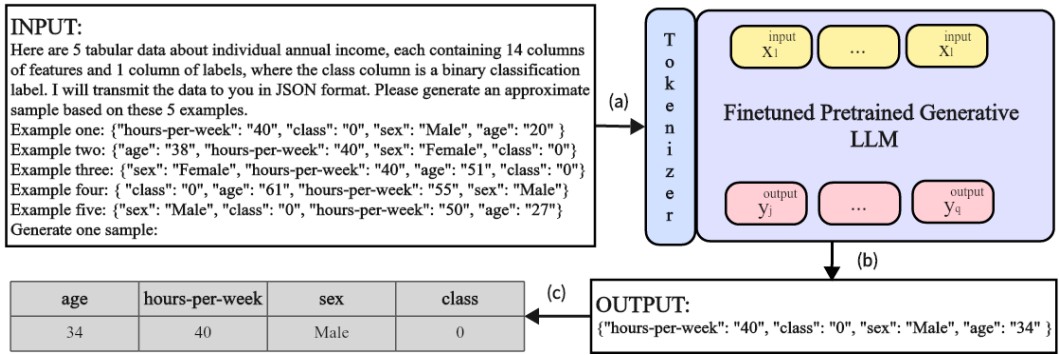

Figure 2: The sampling step. It involves inputting a prompt, shown within the white box in the upper left corner (a), into the fine-tuned pretrained LLM. This results in a textual output (b), which is then converted into a table using pattern matching (c).

significant progress, with the possibility to rival or even surpass state-of-the-art machine learning approaches [36]. Therefore, we propose the idea of using synthetic data to fine-tune a LLM to a classifier and then evaluate such classifier on the original test sets. We refer this evaluation metric as **LLM Effecitveness** (**LLE**). In the current work, we choose LLaMA-2-7b-chat [37] as the base LLM for **LLE**.

### 3.2.2   DLT: Data Leakage Test

The commonly used data leakage metrics Distance to Closest Record (DCR) [10] and NewRowSynthesis (NRS) [32] focus on measuring the "distance" between synthetic data and original data. They do not take into account the extent to which the generator itself leaks the original data. Related research indicates that the LLMs are susceptible to data leakage issues to varying degrees [20]. Attacks on LLMs of synthetic data generator have the potential to extract the complete training data, leading to severe privacy breaches. To address this issue, we propose a new metric for quantifying privacy protection named the **Data Leakage Test (DLT)**, which is inspired by the work of Skywork [38]. This metric measures the level of which a generator leaks original data. The DLT computes the perplexity (ppl) of the generator on both the synthetic and the original data to determine its data generation tendencies.

To compute the DLT, we first feed the training data into the generator to compute the ppl for each sample, then average these scores to determine the ppl on the training data, referred to as ppl-on-train. We then feed the synthetic data into the generator and obtain the ppl, referred to as ppl-on-syn. The DLT value is computed by subtracting ppl-on-syn from ppl-on-train. A larger DLT value indicates better privacy protection of the original data by the generator, whereas a smaller value indicates weaker privacy protection. The formula of DLT is shown as below, where the $P(x)$ denotes the probability of generating a sentence.

$$\text{DLT} = \text{PPL}(\text{D}_{\text{test}}) - \text{PPL}(\text{D}_{\text{train}}) \tag{4}$$

$$\text{PPL}(\text{D}_{\text{split}}) = \frac{1}{|\text{D}_{\text{split}}|} \sum_{x \in \text{D}_{\text{split}}} P(x)^{-\frac{1}{N}} = \frac{1}{|\text{D}_{\text{split}}|} \sum_{x \in \text{D}_{\text{split}}} 2^{\text{Cross}-\text{Entropy(x)}} \tag{5}$$

## 4   Experiment

In this section, we select four real-world datasets to compare the performance of HARMONIC with various types of data synthesis methods. The comparison is conducted in two aspects: the effectiveness of the synthesized data and its privacy protection. [4]

---

[4]Ablation studies of the effectiveness of the kNN operation and other parts of our generation framework are provided in Appendix C. We also assess a synthetic data generation method with differential privacy in Appendix C.

## 4.1 Experimental Setup

**Datasets.** To evaluate the proposed method, we utilize four real-world datasets from various domains, namely *GM* (German [39]), *AD* (Adult Income [40]), *DI* (Diabetes)[5], *BU* (Buddy)[6], which are all open source datasets and don't contain any personal information such as names, phone numbers, addresses, or other sensitive data. These datasets, whose sizes range from fewer than 1,000 to tens of thousands of rows, also differ in feature types and the number of features. Some datasets include only categorical features, while others contain both numerical and categorical features. We divide each dataset into training, validation, and test sets in a ratio of 7:1:2.

**Baselines.** As discussed in related works, we select the most representative methods as our baselines, including: *SMOTE*, a simple interpolation method proposed for oversampling minority classes and can also be used for generating synthetic data [23]; *TVAE*, a state-of-the-art method for tabular data generation based on VAE[9]; *CTABGAN*, a GAN-based model that performs exceptionally well across a diverse set of benchmarks [10]; *TabDDPM*, a famous benchmark for Diffusion-based Methods [14]; *TABSYN*, a faster synthesis compared with other diffusion-based techniques [15]; REaLTabFormer (*RTF*) [24] and *GReaT*, state-of-the-art tabular data synthesizers based on LLMs, to be precise, both are based on GPT-2 [19] [18].

**Metrics.** We evaluate the effectiveness of the synthetic data from two aspects: the statistical distribution characteristics and the effectiveness on the downstream task. For the the statistical distribution characteristics, we employ the metrics "data_mismatch" (*DM*) to assess data type compatibility (0 indicates no datatype mismatch), "Wasserstein_dist" (*WD*) to quantify distributional differences (0 indicates identical distributions) and "CorrelationSimilarity" (*CS*) to evaluate the similarity of column-wise correlations (1 indicates that the pairwise correlations are identical). These are preliminary examinations of the effectiveness of synthetic data in previous work [7]. For further evaluation with effectiveness on the downstream task, we use *MLE*[9] and *LLE* just proposed, which train a machine learning model or a LLM on the original data and the synthetic data and compute the weighted F1 on the test data.

For the ability of privacy protection of the synthetic data, we use three different metrics to evaluate: Distance to Closest Record (DCR) [10] and NewRowSynthesis (NRS) [32], and our proposed *DLT* metric. All three metrics are positively correlated with privacy, meaning that higher values indicate stronger ability of privacy protection.

**Implementation Details.** Our approach allows for the selection of any pre-trained generative LLM that supports fine-tuning, such as GPT-2 [19], LLaMA-2-7b-chat [37], Mistral [41], etc., as the base model. By default, our method choose LLaMA-2-7b-chat [37] as the base model due to its rich pre-training corpus, resulting in a stronger language understanding capability compared with GPT-2 [19]. This enables LLaMA-2-7b-chat [37] to learn fine-tuning tasks more efficiently. Considering the time cost of the entire experiment, we choose LoRA [42] as the efficient fine-tuning method instead of full parameter adjustment. [8]

## 4.2 The Effectiveness of Synthetic Data

**Our method achieves effectiveness comparable to existing state-of-the-art approaches.** In Table 1, DM, WD, CS demonstrate that our method achieves state-of-the-art performance in terms of statistical distribution characteristics of generated data across all synthetic data generation methods, while remaining comparable to other LLM-based approaches in terms of its proximity and correlation between columns to the original distribution. These demonstrate our ability to effectively preserve the original data distribution. Moreover, these results show that our method can capture the relationships between columns, achieving comparable scores with other methods. Notably, on the DI dataset, our method (0.95) significantly outperforms GReaT (0.88), another LLM-based approach. These overall findings further validate the effectiveness of our method.

The performance on the downstream task also demonstrates the effectiveness of our method. Compared with other synthetic methods, our method displays promising results in many scenarios,

---

[5]https://www.openml.org/search?type=data&sort=runs&id=37

[6]https://www.kaggle.com/datasets/akash14/adopt-a-buddy

[7]https://github.com/vanderschaarlab/synthcity

[8]For more details of this section, please refer to Appendix B.

Table 1: The results for effectiveness. The best results are marked in bold, the second-best results are underlined. All results are averages over 3 different random seeds.

| Dataset | Metric | Original | HARMONIC | SMOTE | TVAE | CTAB | TabDDPM | TABSYN | RTF | GReaT |
|---|---|---|---|---|---|---|---|---|---|---|
| GM | DM | – | $\mathbf{0.00_{\pm0.00}}$ | $0.14_{\pm0.00}$ | $0.14_{\pm0.00}$ | $0.14_{\pm0.00}$ | $0.14_{\pm0.00}$ | $0.27_{\pm0.00}$ | $\mathbf{0.00_{\pm0.00}}$ | $0.14_{\pm0.00}$ |
|  | WD | – | $0.87_{\pm0.07}$ | $0.85_{\pm0.03}$ | $\underline{0.70_{\pm0.04}}$ | $0.77_{\pm0.02}$ | $0.73_{\pm0.02}$ | $0.94_{\pm0.09}$ | $\mathbf{0.67_{\pm0.01}}$ | $0.93_{\pm0.22}$ |
|  | CS | – | $0.96$ | $0.97$ | $\mathbf{0.99}$ | $\underline{0.98}$ | $0.90$ | $\underline{0.98}$ | $\underline{0.98}$ | $\underline{0.98}$ |
|  | MLE | $0.50_{\pm0.00}$ | $0.55_{\pm0.03}$ | $0.64_{\pm0.02}$ | $0.61_{\pm0.02}$ | $0.57_{\pm0.02}$ | $0.64_{\pm0.01}$ | $0.63_{\pm0.02}$ | $\mathbf{0.65_{\pm0.01}}$ | $0.44_{\pm0.03}$ |
|  | LLE | $0.71_{\pm0.00}$ | $0.64_{\pm0.03}$ | $\underline{0.67_{\pm0.04}}$ | $0.69_{\pm0.03}$ | $0.71_{\pm0.04}$ | $0.67_{\pm0.05}$ | $\mathbf{0.72_{\pm0.02}}$ | $0.69_{\pm0.03}$ | $0.55_{\pm0.11}$ |
| AD | DM | – | $0.21_{\pm0.15}$ | $\mathbf{0.00_{\pm0.00}}$ | $\mathbf{0.00_{\pm0.00}}$ | $\mathbf{0.00_{\pm0.00}}$ | $\mathbf{0.00_{\pm0.00}}$ | $\mathbf{0.00_{\pm0.00}}$ | $\mathbf{0.00_{\pm0.00}}$ | $0.06_{\pm0.00}$ |
|  | WD | – | $0.48_{\pm0.15}$ | $0.49_{\pm0.01}$ | $0.31_{\pm0.04}$ | $0.07_{\pm0.01}$ | $\underline{0.06_{\pm0.01}}$ | $0.07_{\pm0.01}$ | $\mathbf{0.03_{\pm0.00}}$ | $3.83_{\pm0.09}$ |
|  | CS | – | $0.90$ | $\mathbf{0.99}$ | $\underline{0.98}$ | $0.97$ | $\mathbf{0.99}$ | $0.97$ | $\mathbf{0.99}$ | $0.94$ |
|  | MLE | $0.61_{\pm0.00}$ | $0.67_{\pm0.02}$ | $0.75_{\pm0.00}$ | $0.74_{\pm0.00}$ | $0.73_{\pm0.01}$ | $0.74_{\pm0.00}$ | $0.73_{\pm0.00}$ | $\mathbf{0.76_{\pm0.00}}$ | $0.73_{\pm0.01}$ |
|  | LLE | $0.81_{\pm0.00}$ | $0.80_{\pm0.02}$ | $\underline{0.84_{\pm0.01}}$ | $0.83_{\pm0.01}$ | $0.83_{\pm0.00}$ | $0.83_{\pm0.00}$ | $0.81_{\pm0.02}$ | $\mathbf{0.85_{\pm0.00}}$ | $0.82_{\pm0.02}$ |
| DI | DM | – | $\mathbf{0.03_{\pm0.05}}$ | $\underline{0.07_{\pm0.05}}$ | $0.10_{\pm0.00}$ | $0.10_{\pm0.00}$ | $0.07_{\pm0.05}$ | $0.10_{\pm0.00}$ | $0.10_{\pm0.00}$ | $0.07_{\pm0.05}$ |
|  | WD | – | $0.14_{\pm0.01}$ | $\mathbf{0.07_{\pm0.00}}$ | $\mathbf{0.07_{\pm0.00}}$ | $0.26_{\pm0.01}$ | $\underline{0.08_{\pm0.00}}$ | $0.09_{\pm0.01}$ | $0.09_{\pm0.02}$ | $0.13_{\pm0.00}$ |
|  | CS | – | $0.95$ | $0.96$ | $\underline{0.98}$ | $0.97$ | $\mathbf{0.99}$ | $\underline{0.98}$ | $\mathbf{0.99}$ | $0.88$ |
|  | MLE | $0.56_{\pm0.00}$ | $0.46_{\pm0.02}$ | $\mathbf{0.72_{\pm0.03}}$ | $0.71_{\pm0.02}$ | $0.67_{\pm0.00}$ | $0.71_{\pm0.02}$ | $0.68_{\pm0.03}$ | $0.66_{\pm0.03}$ | $0.45_{\pm0.03}$ |
|  | LLE | $0.70_{\pm0.00}$ | $0.75_{\pm0.00}$ | $0.69_{\pm0.04}$ | $\underline{0.72_{\pm0.04}}$ | $0.62_{\pm0.09}$ | $0.72_{\pm0.03}$ | $\mathbf{0.77_{\pm0.01}}$ | $0.70_{\pm0.04}$ | $0.71_{\pm0.03}$ |
| BU | DM | – | $\mathbf{0.00_{\pm0.00}}$ | $0.00_{\pm0.00}$ | $0.00_{\pm0.00}$ | $0.00_{\pm0.00}$ | $0.00_{\pm0.00}$ | $0.00_{\pm0.00}$ | $0.00_{\pm0.00}$ | $0.03_{\pm0.04}$ |
|  | WD | – | $0.48_{\pm0.16}$ | $0.23_{\pm0.02}$ | $0.10_{\pm0.01}$ | $\underline{0.05_{\pm0.00}}$ | $0.06_{\pm0.00}$ | $\mathbf{0.04_{\pm0.00}}$ | $\mathbf{0.04_{\pm0.00}}$ | $2292.47_{\pm1014.22}$ |
|  | CS | – | $0.93$ | $0.98$ | $0.97$ | $\underline{0.99}$ | $\mathbf{1.00}$ | $\mathbf{1.00}$ | $0.98$ | $\mathbf{1.00}$ |
|  | MLE | $0.38_{\pm0.00}$ | $\mathbf{0.27_{\pm0.03}}$ | $0.25_{\pm0.02}$ | $\mathbf{0.27_{\pm0.03}}$ | $0.26_{\pm0.01}$ | $\mathbf{0.27_{\pm0.01}}$ | $0.26_{\pm0.01}$ | $0.26_{\pm0.00}$ | $0.24_{\pm0.03}$ |
|  | LLE | $0.88_{\pm0.00}$ | $0.82_{\pm0.03}$ | $0.85_{\pm0.04}$ | $\mathbf{0.86_{\pm0.01}}$ | $0.82_{\pm0.02}$ | $\underline{0.85_{\pm0.01}}$ | $\mathbf{0.86_{\pm0.01}}$ | $0.70_{\pm0.14}$ | $0.81_{\pm0.03}$ |

particularly considering privacy protection in the subsequent section, resulting in a more balanced solution. While our method only achieves the best performance on the MLE metric of the BU dataset, it exhibits comparable results to current state-of-the-art generative methods in other datasets. This indicates that our data synthesis method is effective and performs on par with existing approaches. Even when prioritizing data leakage protection, our method may be a better choice. Compared with the original data, our method surpasses the original training set on the DI dataset, and on the remaining three datasets our performance only falls slightly short. The average decrease compared with the original data benchmark is less than 5%, which falls within an acceptable range for practical applications.

In conclusion, while our method may not achieve the absolute highest performance on every dataset, the results presented in this section overwhelmingly support its potential as a viable substitute for original data. The synthetic data generated by our method demonstrates both effectiveness and stability, making it a valuable tool for various LLM-based applications.

**Relying solely on MLE metrics may lead to inaccurate conclusions, and LLE is an important potential metric for synthetic data evaluation.** By examining our LLE metric, we observe that the MLE metric alone may not comprehensively reflect the effectiveness of different synthetic datasets. The evaluation results of LLE and MLE are not always consistent for the same method. For instance, TABSYN often performs better on LLE, while RTF excels on the MLE metric. This suggests that different synthetic data methods may have varying levels of effectiveness for downstream models (MLE and LLE), and a single evaluation metric may not adequately capture the true impact of a synthetic data approach.

More importantly, LLE highlights the potential of LLMs in utilizing synthetic tabular data, potentially surpassing traditional machine learning methods. This is particularly evident in the BU dataset. This finding suggests that leveraging LLMs to better accomplish tasks through synthetic data is a promising future direction. Therefore, the LLE metric holds significant potential in measuring the effectiveness of synthetic data.

## 4.3 The Privacy of Synthetic Data

**The experimental results demonstrate that our method prioritizes privacy in the synthetic data generation.** This is particularly beneficial in situations where disclosing original data is not feasible due to privacy concerns. In such scenarios, our synthetic data serves as a reliable and secure substitute for original data, allowing downstream tasks to proceed without compromising sensitive information.

Table 2 presents three key privacy metric scores to quantify the privacy protection of our method. Analyzing the results in Table 2, it's evident that our method surpasses or comes in a close second for almost all datasets across all three metrics. This translates to demonstrably stronger privacy protection compared with existing methods.

Table 2: The results for privacy. The best results are marked in bold, the second-best results are underlined. Each dataset has three metrics, and in all cases, higher values are better.

| Dataset | Metric | HARMONIC | SMOTE | TVAE | CTAB | TabDDPM | TABSYN | RTF | GReaT |
|---------|--------|----------|-------|------|------|---------|--------|------|-------|
| GM | NRS | **1.00** | 1.00 | 1.00 | 1.00 | 1.00 | 1.00 | 1.00 | 1.00 |
|  | DCR | **8.08** | 2.77 | 4.09 | 5.36 | 2.21 | 3.98 | 4.60 | 5.84 |
|  | DLT | **-0.16** | — | — | — | — | — | -22.04 | -2.14 |
| AD | NRS | **1.00** | 0.95 | 1.00 | 1.00 | 1.00 | 1.00 | 1.00 | 1.00 |
|  | DCR | **2.47** | 0.16 | 0.49 | 0.82 | 0.50 | 0.86 | 0.57 | 1.51 |
|  | DLT | -0.98 | — | — | — | — | — | -163.71 | **-0.67** |
| DI | NRS | **1.00** | 1.00 | 1.00 | 1.00 | 1.00 | 1.00 | 1.00 | 1.00 |
|  | DCR | 0.44 | 0.28 | 0.33 | 0.72 | 0.21 | **1.37** | 0.36 | 1.36 |
|  | DLT | **-0.37** | — | — | — | — | — | -42.46 | -0.44 |
| BU | NRS | **1.00** | 0.93 | 1.00 | 1.00 | 0.99 | 1.00 | 1.00 | 1.00 |
|  | DCR | 2.52 | 0.15 | 0.66 | 0.70 | 0.18 | 1.38 | 0.38 | **8.30** |
|  | DLT | **-0.34** | — | — | — | — | — | -41.13 | -2.22 |

Moreover, besides the metrics, the design of our method inherently offers superior security in practice. If an attacker attempts to reconstruct a row of original data, he/she needs to know nearly the $k$ rows of original data first. This includes knowing the sequence of each feature within a record and the specific order of these $k$ samples. This significantly raises the bar for attackers compared with methods like GReaT, which exposes a vulnerability where an attacker with knowledge of just one or two feature values in an original record can potentially reconstruct the entire record.

## 5 Conclusion

In this paper, we introduce HARMONIC, a novel framework that leverages the power of LLMs for synthesizing tabular data but taking privacy concerns into account. It enables LLMs to capture both the internal feature relationships within individual row of data point and the broader connections among data point by instruction fine-tuning which is the key for the improvement in privacy protection. We also propose the metric named DLT specifically for evaluating the level of privacy protection in the synthetic data by LLM. Extensive evaluations across four real-world datasets of classification tasks showcase the ability of HARMONIC in the crucial balance of effectiveness and privacy protection.

**Limitations and Future Work**. We conclude the paper with the limitations and future work: (1) Compared with other GPT-2 based methods, our approach requires a longer processing time for larger LLMs. However, take a step forward, we believe our method will become more applicable to a wider range of contexts as hardware performance improves and cloud computing advances. (2) LLMs are less sensitive to numerical data and are better suited for classification tasks rather than regression tasks. As a result, our current work focuses primarily on tabular data used for classification tasks.[9] (3) It would be interesting to carry out the the comparison and the integration of differential privacy with our method which may be a focus of our future work. (4) The ethical and potential biases of synthetic data are also critical concerns. Generating the synthetic data can inadvertently perpetuate existing biases. Addressing this challenge remains an open problem in the area of data synthesis.

## Acknowledgements

We would like to thank the editors and reviewers for their insightful comments and guidance, which significantly improved this work. This research is supported by the National Key R&D Program of China (No. 2022YFC3301503) and Sichuan Key Laboratory of AI Empowered Governance in Smart Society, China. It is also supported by the project JPNP20006 from New Energy and Industrial Technology Development Organization (NEDO), Artificial Intelligence Research Center, National Institute of Advanced Industrial Science and Technology, Japan, National Science and Technology Major Project (No.2021ZD0113304), National Natural Science Foundation of China (U23A20316), and Joint&Laboratory on Credit Technology. The views expressed in this paper are solely those of the author and do not necessarily reflect the views of their affiliated institutions or funding organizations.

---

[9]We also have supplementary experiments to explore regression task in the Appendix C.

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

# A  Datasets Details

## A.1  Data Source

We list the sources of our datasets in Table 3, all of which are obtained from publicly accessible and reputable websites.

Table 3: URLs for real-world datasets of the experiments.

| Dataset | URL |
|---|---|
| German | `https://archive.ics.uci.edu/dataset/144/statlog+german+credit+data` |
| Adult Income | `https://archive.ics.uci.edu/dataset/2/adult` |
| Diabetes | `https://www.openml.org/search?type=data&sort=runs&id=37&status=active` |
| Buddy | `https://www.kaggle.com/datasets/akash14/adopt-a-buddy` |

## A.2  Data Description

Additionally, we record various statistical details for each dataset in Table 4.

**German.** The German dataset classifies people as good or bad credit risks described by a set of attributes including status of existing checking account, duration in month, credit history, purpose and more.

**Adult Income.** The US Adult income dataset was extracted by Barry Becker from the 1994 US Census Database. The dataset consists of anonymous information such as occupation, age, native country, race, capital gain, capital loss, education, work class and more. Each row is labelled as either having a salary greater than ">50K" or "<=50K".

**Diabetes.** The Diabetes dataset originates from the National Institute of Diabetes and Digestive and Kidney Diseases. This dataset comprises medical features including the number of times pregnant, diastolic blood pressure, body mass index, age, among other variables. The label indicates whether the individual has diabetes or not.

**Buddy.** The Buddy dataset originates from the HackerEarth Machine Learning Challenge—Adopt a Buddy. The dataset consists of parameters such as: a unique ID assigned to each animal that is up for adoption, date on which they arrived at the shelter, their physical attributes such as color, length and height, among other factors. The labels in this dataset denote the breed of the animals.

Table 4: Dataset Statistics. # Samples denotes the number of samples in each dataset. # Num and # Cat columns indicate numbers of numerical and categorical features in each dataset.

| Dataset | Domain | # Samples | # Num | # Cat | Tasks | # Classes |
|---|---|---|---|---|---|---|
| German | Financial | 1000 | 7 | 13 | Classification | 2 |
| Adult Income | Social | 32561 | 6 | 8 | Classification | 2 |
| Diabetes | Medical | 768 | 8 | 0 | Classification | 2 |
| Buddy | Nature | 18834 | 4 | 5 | Multi-Class | 3 |

## A.3  Data Preprocessing

To maintain consistency in formatting, we converted all four datasets into CSV files. Additionally, the other datasets underwent the following preprocessing steps:

**German.** The original label "status" with a value of "1" was converted to "0", and the original label "status" with a value of "2" was converted to "1".

**Adult Income.** The original label "class" with a value of "<=50K" was converted to "0", and the original label "class" with a value of ">50K" was converted to "1".

**Diabetes.** The diabetes dataset was used without any additional preprocessing.

**Buddy.** The original "issue_date" and "listing_date," which were represented in the "date_time" format, have been replaced with a timestamp format.

### A.4 Data Field

The instruction fine-tunnig dataset is provided in json format and contains the following attributes. And a specific instance of INPUT and OUTPUT can be found in A.5.

```
{
    id: [integer] The unique identifier for each instance
    conversations: [
        {
            from: [string] "human"
            value: [string] the INPUT text for LLM fine-tuning
        },
        {
            from: [string] "assistant"
            value: [string] the OUTPUT text for LLM fine-tunnig
        }
    ]
}
```

### A.5 Data Instance

To illustrate the data format used for fine-tuning both the generator and downstream tasks, we present a complete data instance from the German dataset as an example, shown in Table 5 and Table 6 respectively.

## B  Experimental Details

### B.1 Parameter Selection

Considering the time cost of the entire experiment, we did not adjust the best hyperparameters for different dataset. By conducting experiments on the validation set and combining empirical settings, we unified the hyperparameters of the fine-tuning process. In the fine-tuning stage, we choose lora[42] efficient fine-tuning instead of full parameter adjustment.

We fine-tune the LLaMA-2-7b-chat model for each dataset for 5 epochs with a batch size of 16. We utilize the AdamW optimizer for the proposed generative models, with the learning rate $3 \times 10^{-4}$.

For the sampling step, we use 3 random seeds in the data generation stage for each dataset, specifically 1234, 1235, and 1236. We set the temperature parameter T to 0.7 for all experiments and datasets. We sample new synthetic data using the prompt dataset for generation (Sec 3.1.1), starting with task description and five random original samples(see an example in Appendix A.5). We generated synthetic datasets for German and Diabetes with the same number of samples as their respective training sets. For the Adult Income and Buddy datasets, where the training sets are larger, exceeding 10,000 samples, we generated 5,000 samples due to the extended time required for sampling with our method.

For the MLE metric, we employ logistic regression, decision tree, mlp and random forest models.

For the LLE metric, the epoch set for fine-tuning the downstream LLaMA-2-7b-chat model is 5, the learning rate is $1 \times 10^{-4}$, and the batch size is 32. The random seed is fixed when fine-tuning the downstream model. See an example of instruction data for downstream tasks in Appendix A.5).

### B.2 Experimental Environment

Our hardware setup includes 4 NVIDIA A100-40GB GPUs. The system has 1 TB system RAM, and runs on an AMD EPYC 7742 processor with 64 cores, using the Ubuntu 22.04 operating system.

Table 5: An instance of the instruction data fine-tuning for the generator training.

**INPUT:** Here are 5 tabular data about user credit scores, each containing 20 columns of features and 1 column of labels, where the 'status' column is a binary classification label. I will transmit the data to you in JSON format. Please generate an approximate sample based on these 5 examples.\n Example one: {"Present employment since": "A75", "Credit amount": "11816", "Credit history": "A30", "Purpose": "A49", "Duration in month": "45", "Other installment plans": "A143", "Age in years": "29", "Savings account/bonds": "A61", "status": "1", "foreign worker": "A201", "Number of people being liable to provide maintenance for": "1", "Number of existing credits at this bank": "2", "Installment rate in percentage of disposable income": "2", "Housing": "A151", "Property": "A123", "Present residence since": "4", "Telephone": "A191", "Other debtors / guarantors": "A101", "Job": "A173", "Status of existing checking account": "A11", "Personal status and sex": "A93"}.\n Example two: {"Housing": "A151", "Personal status and sex": "A92", "Credit amount": "6416", "Job": "A173", "Property": "A124", "Purpose": "A49", "status": "1", "Number of people being liable to provide maintenance for": "1", "Number of existing credits at this bank": "1", "Present employment since": "A75", "Other installment plans": "A143", "Installment rate in percentage of disposable income": "4", "Present residence since": "3", "Status of existing checking account": "A12", "Savings account/bonds": "A61", "Telephone": "A191", "Other debtors / guarantors": "A101", "Age in years": "59", "Duration in month": "48", "Credit history": "A31", "foreign worker": "A201"}.\n Example three: {"Housing": "A151", "Installment rate in percentage of disposable income": "4", "Age in years": "31", "Duration in month": "24", "foreign worker": "A201", "Number of people being liable to provide maintenance for": "1", "Other installment plans": "A143", "Savings account/bonds": "A61", "Present employment since": "A73", "Credit history": "A31", "Status of existing checking account": "A11", "Job": "A173", "Telephone": "A192", "Number of existing credits at this bank": "1", "status": "1", "Personal status and sex": "A93", "Credit amount": "3161", "Other debtors / guarantors": "A101", "Purpose": "A49", "Property": "A122", "Present residence since": "2"}.\n Example four: {"Purpose": "A49", "Number of people being liable to provide maintenance for": "1", "Housing": "A151", "Age in years": "26", "Savings account/bonds": "A62", "Other installment plans": "A143", "Present employment since": "A73", "Telephone": "A191", "Installment rate in percentage of disposable income": "4", "Duration in month": "30", "Number of existing credits at this bank": "2", "Personal status and sex": "A92", "Present residence since": "4", "Status of existing checking account": "A12", "Job": "A172", "Credit history": "A30", "Property": "A123", "Other debtors / guarantors": "A101", "status": "1", "Credit amount": "4280", "foreign worker": "A201"}.\n Example five: {"Present employment since": "A74", "Credit amount": "3566", "Duration in month": "48", "foreign worker": "A201", "Other debtors / guarantors": "A101", "Other installment plans": "A143", "Number of existing credits at this bank": "1", "Number of people being liable to provide maintenance for": "1", "Credit history": "A31", "Housing": "A152", "Present residence since": "2", "Installment rate in percentage of disposable income": "4", "Savings account/bonds": "A62", "Telephone": "A191", "status": "0", "Job": "A173", "Purpose": "A49", "Age in years": "30", "Personal status and sex": "A93", "Property": "A123", "Status of existing checking account": "A12"}.\n Generate one sample:

**OUTPUT:** {"Present residence since": "4", "Credit amount": "7685", "Age in years": "37", "Other installment plans": "A143", "Status of existing checking account": "A11", "Housing": "A151", "Credit history": "A31", "Duration in month": "48", "Property": "A123", "Purpose": "A49", "Other debtors / guarantors": "A103", "Present employment since": "A74", "Installment rate in percentage of disposable income": "2", "Job": "A173", "Savings account/bonds": "A61", "Telephone": "A191", "Number of people being liable to provide maintenance for": "1", "Number of existing credits at this bank": "1", "Personal status and sex": "A92", "foreign worker": "A201", "status": "1"}.

## C  Additional results

The following presents the results of the ablation study. We conducted comparative experiments using the German and Diabetes datasets.

### C.1  Filter operation

Experimental results demonstrate that the filtering step in our generation framework (the last step in the **Construct the instruction fine-tuning dataset using kNN** of section 3.1.1) can enhance the effectiveness of synthetic data. As shown in Table 7, the LLE values decrease without filtering, particularly for the German dataset. This is likely due to incorrect labels in the generated synthetic data. Additionally, privacy slightly diminishes without the filtering step, though the difference is minimal. These findings indicate that the filtering step is effective.

Table 6: An instance of the prompt dataset for tabular data synthesis.

| |
|---|
| **INPUT:** Evaluate the creditworthiness of a customer with the following financial profile. Respond with only either 'good' or 'bad'. \n Text: 'The state of Status of existing checking account is bigger than 0 DM but smaller than 200 DM, The state of Duration in month is 36, The state of Credit history is delay in paying off in the past, The state of Purpose is car (new), The state of Credit amount is 1873, The state of Savings account or bonds is bigger than 100 smaller than 500 DM, The state of Present employment since is bigger than 1 smaller than 4 years, The state of Installment rate in percentage of disposable income is 2, The state of Personal status and sex is male and single, The state of Other debtors or guarantors is none, The state of Present residence since is 2, The state of Property is unknown or no property, The state of Age in years is 29, The state of Other installment plans is none, The state of Housing is for free, The state of Number of existing credits at this bank is 1.0, The state of Job is management or self-employed or highly qualified employee or officer, The state of Number of people being liable to provide maintenance for is 1, The state of Telephone is yes, registered under the customers name, The state of foreign worker is yes.'\n Answer: |
| **OUTPUT:** "bad" |

Table 7: The results of whether to filter data after kNN, where "w/o fil" means not to filter data, and "with fil" means to filter data, which is our original method. Each dataset has five metrics, and in all cases, higher values are better.

| Dataset | Filter | MLE | LLE | NRS | DCR | DLT |
|---|---|---|---|---|---|---|
| GM | w/o fil | $0.56_{\pm0.06}$ | $0.59_{\pm0.03}$ | 1.00 | 7.97 | -0.17 |
| | with fil | $0.55_{\pm0.03}$ | $\mathbf{0.64_{\pm0.03}}$ | 1.00 | **8.08** | **-0.16** |
| DI | w/o fil | $0.56_{\pm0.06}$ | $0.74_{\pm0.01}$ | 1.00 | 0.44 | -0.38 |
| | with fil | $0.46_{\pm0.02}$ | $\mathbf{0.75_{\pm0.00}}$ | 1.00 | 0.44 | **-0.37** |

## C.2 Random feature order permutation

Our experiments' results indicate that the permuting features in our generation framework (the last step in the **Data format conversion** of section 3.1.1) can enhance the privacy of synthetic data. As shown in the last two columns of Table 8, there is a significant reduction in both the DCR and DLT values when features are not permuted. Concurrently, the generated numerical columns tend to produce repeated values, which may also contribute to the decrease in the LLE metric. Overall, these results underscore the necessity of shuffling features.

Table 8: The results of whether to shuffle features, where "w/o pm" means not to shuffle the features, and "with pm" means to shuffle the features, which is our original method. Each dataset has five metrics, and in all cases, higher values are better.

| Dataset | Permutation | MLE | LLE | NRS | DCR | DLT |
|---|---|---|---|---|---|---|
| GM | w/o pm | $0.56_{\pm0.04}$ | $0.63_{\pm0.05}$ | 1.00 | 7.20 | -0.58 |
| | with pm | $0.55_{\pm0.03}$ | $\mathbf{0.64_{\pm0.03}}$ | 1.00 | **8.08** | **-0.16** |
| DI | w/o pm | $0.50_{\pm0.06}$ | $0.70_{\pm0.03}$ | 1.00 | 0.42 | -0.67 |
| | with pm | $0.46_{\pm0.02}$ | $\mathbf{0.75_{\pm0.00}}$ | 1.00 | **0.44** | **-0.37** |

## C.3 Compare with PrivBayes

We supplement a experiment of the PrivBayes data synthesis method on the Diabetes dataset to explore the effect of data leakage protection of our method compared with the traditional differential privacy method.

As shown in Table 9, our method offers a better balance between data leakage protection and data effectiveness compared with differential privacy methods. Although our method exhibits a lower DCR (privacy metric) compared with PrivBayes with differential privacy enabled, our approach consistently outperforms PrivBayes in DM, WD and CS (statistical distribution characteristics) and LLE (the utility of the synthetic data). We also achieve near-identical results in MLE. This is because differential privacy prioritizes strong privacy guarantees, often at the expense of performance. However, our method significantly improve the privacy compared with other existing approaches

without compromising effectiveness. Moreover, our method does not conflict with differential privacy approaches and can also combines with differential privacy for further data leakage protection.

Table 9: Compare with PrivBayes on Diabetes. In all cases, higher values are better.

| Method | DM | WD | CS | MLE | LLE | NRS | DCR |
|---|---|---|---|---|---|---|---|
| PrivBayes | $0.10_{\pm 0.00}$ | $0.34_{\pm 0.01}$ | 0.88 | $0.48_{\pm 0.04}$ | $0.20_{\pm 0.02}$ | 1.00 | 0.82 |
| HARMONIC | $\mathbf{0.03_{\pm 0.05}}$ | $\mathbf{0.14_{\pm 0.01}}$ | **0.95** | $0.46_{\pm 0.02}$ | $\mathbf{0.75_{\pm 0.00}}$ | **1.00** | 0.44 |

## C.4 Random sampling

To assess the impact of the kNN approach employed for constructing the instruction-tuning dataset, we perform ablation studies on the German and Diabetes datasets. A control group is established where 5 data points were randomly sampled from the training set, instead of utilizing kNN to identify 5 nearest neighbors for each data point. This variation facilitate the extraction of data points that were far from the target sample. All other experimental configurations and processes remain consistent with those outlined in the main text.

Table 10: The results of whether to use kNN, where "random" means random sampling, and "kNN" means using kNN, which is our original method. Each dataset has five metrics, and in all cases, higher values are better.

| Dataset | Method | MLE | LLE | NRS | DCR | DLT |
|---|---|---|---|---|---|---|
| GM | random | $0.42_{\pm 0.00}$ | $0.49_{\pm 0.04}$ | 1.00 | 4.00 | -0.14 |
| | kNN | $\mathbf{0.55_{\pm 0.03}}$ | $\mathbf{0.64_{\pm 0.03}}$ | 1.00 | **8.08** | -0.16 |
| DI | random | $0.61_{\pm 0.06}$ | $0.70_{\pm 0.05}$ | 0.98 | 0.62 | -0.36 |
| | kNN | $0.46_{\pm 0.02}$ | $\mathbf{0.75_{\pm 0.00}}$ | **1.00** | 0.44 | -0.37 |

The results in Table 10 demonstrate that data synthesis using random sampling consistently underperforms our KNN-based approach, with an average decrease of 23% in performance on German dataset. On the Diabetes dataset, our KNN-based approach also performs better on the LLE metric. This suggests that LLMs, when synthesizing data, still require a degree of similarity and context to effectively learn the relationships between data points. Random sampling, by introducing less relevant data, may hinder the LLM's ability to capture these relationships.

## C.5 Regrssion task

We expand a regression dataset, Abalone[10] for our experiments. In the regression task, the MLE and LLE metrics are evaluated using R2 score rather than F1 score. All other aspects of the experimental framework, including procedures and configurations, remain identical to those in the main text.

The experimental results in Table 11 demonstrate that our method not only maintains strong privacy protection but also have potential in regression tasks compared with other LLM-based tabular synthetic methods. We achieve a higher DCR (0.18 vs. 0.15 for GReaT and 0.11 for RTF) and a higher DLT (-0.38 vs. -0.71 for GReaT and -68.99 for RTF), signifying our ability to maintain privacy even for regression tasks. Similarly, our method displays consistent performance in terms of data effectiveness, measured by LLE and MLE metrics, while RTF exhibits significantly worse performance in LLE (-32.67), and GReaT experiences a considerable decrease in MLE (0.09).

## D Ethics Statement

The dataset used in this study is based on open-source data and can be further modified. We thoroughly reviewed and verified the data to ensure it does not contain any personally identifiable information or offensive content. Additionally, we conducted manual audits to ensure there are no sensitive details. Therefore, we believe the dataset is secure and its use in the research is ethically sound and appropriate for the purposes of this study.

---

[10]https://www.openml.org/search?type=data&sort=runs&id=183&status=active

Table 11: The results on Abalone dataset. The best results are marked in bold, the second-best results are underlined. All results are averages over 3 different random seeds.

| Dataset | Metric | Original | HARMONIC | SMOTE | TVAE | CTAB | TabDDPM | TABSYN | RTF | GReaT |
|---------|--------|----------|----------|-------|------|------|---------|--------|-----|-------|
| AB | MLE | $0.42_{\pm 0.00}$ | $0.24_{\pm 0.02}$ | $\mathbf{0.40}_{\pm \mathbf{0.01}}$ | $0.22_{\pm 0.03}$ | – | $\underline{0.35}_{\pm 0.02}$ | $0.33_{\pm 0.01}$ | $0.33_{\pm 0.02}$ | $0.09_{\pm 0.04}$ |
| | LLE | $0.39_{\pm 0.00}$ | $0.22_{\pm 0.03}$ | $\underline{0.25}_{\pm 0.03}$ | $0.00_{\pm 0.17}$ | – | $0.12_{\pm 0.18}$ | $0.21_{\pm 0.10}$ | $-32.67_{\pm 46.36}$ | $\mathbf{0.27}_{\pm \mathbf{0.03}}$ |
| | NRS | – | **1.00** | 0.88 | 1.00 | – | 1.00 | 1.00 | 1.00 | 1.00 |
| | DCR | – | **0.18** | 0.05 | **0.18** | – | 0.14 | 0.13 | 0.11 | 0.15 |
| | DLT | – | **-0.38** | – | – | – | – | – | -68.99 | −0.71 |

