# OpenReview forum: "HARMONIC: Harnessing LLMs for Tabular Data Synthesis and Privacy Protection"
_NeurIPS.cc/2024/Datasets_and_Benchmarks_Track — NeurIPS 2024 Track Datasets and Benchmarks Poster_

### Official Review · Reviewer_N5C1 · 2024-06-14
**Use LLMs for private tabular data synthesis**

**Rating:** 7
**Confidence:** 4
**Correctness:** Yes.

**Review:**

This paper addresses an interesting problem: synthetic tabular data generation with privacy protection. It is well-written, and the method is easy to follow. Using KNN to generate the fine-tuning dataset is an interesting idea, which is simple yet effective. However, this paper does not discuss some related work that focuses on generating datasets with differential privacy.

**Strengths:**

1. This paper addresses an important problem: how to generate private synthetic data using LLMs. Sensitive data, such as medical records and credit information, requires protection and is not widely available. Private synthetic data can prevent the leakage of sensitive information while allowing valuable data to be shared for training models.

2. This paper uses a novel KNN to construct instruction fine-tune dataset, which helps the LLMs learn the distribution of original data. The method is easy to implement yet efficient.

3. This paper proposes two new metrics that measure the accuracy and privacy of the generated dataset.

**Additional Feedback:**

Typo on page 5, line 157, "largere" should be "larger"

**Clarity:**

This paper is well-written and easy to understand, with a clear and logical flow.

**Documentation:**

The github repo provided is not accessible.

**Ethics:**

No.

**Limitations:**

1. The proposed PPL metric seems only work for LLM-related methods, which is limited.

2. There is a line of work focused on generating synthetic data with differential privacy, such as RAP++, DP-CTGAN, and the recent work available at https://arxiv.org/pdf/2406.01457. The author does not discuss these studies as related work.

**Opportunities For Improvement:**

1. In the fine-tuning dataset, only records that are close to the sample are chosen. Would it be helpful to also include records that are far from the sample? This could potentially help LLMs learn the data distribution more comprehensively.

2. Learning the correlation between columns may help improve utility. The current method does not guarantee that the correlations between columns are maintained, which could be a limitation.

3. There is no formal garantee of privacy, I doubt that with proper engineering, membership inference attack can successfully guess whether a certain record belongs to the original dataset.

**Relation To Prior Work:**

This paper lacks a discussion on differentially private synthetic data generation, which offers rigorous privacy guarantees while maintaining the utility of downstream learning tasks.

**Summary And Contributions:**

This paper introduces HARMONIC, an LLM-based framework for private synthetic data generation. Unlike previous approaches that did not prioritize data privacy, HARMONIC allows LLMs to randomly sample synthetic data from a hidden space learned from an instruction fine-tuning dataset. To ensure accuracy, the generated data shares a similar distribution with the original data, maintaining utility for downstream learning tasks. For privacy, the original data is protected because sensitive information is not directly presented. The authors also propose two metrics to measure both accuracy and privacy.

---

> ### Author Rebuttal · Authors · 2024-08-16
>
> Thank you for acknowledging the contribution and writing of our paper, and for your insightful feedback. We will address each of your points below and hope that these responses adequately address your concerns.
>
> **Question 1: Discuss more related work (lack differentially private method)**
>
> Thank you for your suggestion. We will incorporate two key improvements based on your feedback in the revision.
>
> Firstly, we will include a discussion of differentially private methods in our related work section for a comprehensive review and comparison. It's important to note that our method is not in conflict with differentially private methods, and can even be further integrated with them to achieve stronger privacy guarantees. While these differentially private methods offer a path towards privacy preservation on top of existing data synthesis methods, our method directly modifies the data synthesis method itself to enhance privacy. This means our method can be further integrated with differentially private techniques, and we will discuss this potential in the future work section.
>
> Secondly, as also suggested by Reviewer YZek, we will conduct some comparative experiments like PrivBayes. We will provide updated results and analysis throughout this discussion round and in the final revision.
>
> **Question 2:  PPL metric only work for LLM**
>
> Thank you for your feedback. We want to clarify this potential misunderstanding. The PPL-based privacy evaluation metric we utilize is specifically designed for LLMs (lines 50-52). This is due to the fact that while LLMs have demonstrated remarkable data synthesis capabilities in past research, they also pose a significant risk of data leakage (lines 35-40). Existing privacy leakage detection metrics in the synthetic data field might not be suitable for evaluating LLM-based methods (lines 212-216). Therefore, we have specifically constructed this PPL-based metric to address this challenge, which is also a major contribution of our work.
>
> Based your feedback, we will further emphasize this point in the introduction in the revised version.
>
> **Question 3: Github link invalid & Typoes**
>
> Thank you for pointing out this oversight. We had previously set the project to private, but it is now public and available at https://github.com/Wendy619/HARMONIC. In addition, we will address all the typos in the final revision.
>
> **Suggestion 1: Ablation study of records that are far from the sample**
>
> Thank you for your suggestion. Following your comment, we have conducted additional experiments, the results of which are presented in the following Table. We will incorporate these findings in the revised version.
>
> To assess the impact of the kNN approach employed for constructing the instruction-tuning dataset, we performed ablation studies on the German and Diabetes datasets. A control group was established where, instead of utilizing kNN to identify 5 nearest neighbors for each data point, 5 data points were randomly sampled from the training set. This variation facilitated the extraction of data points that were far from the target sample. All other experimental configurations and processes remained consistent with those outlined in the main text.
>
> |Data|Method|MLE|LLE|NRS|DCR|DLT|
> |--|--|---|--|--|--|--|
> |German|random|0.42±0.00|0.49±0.04|1.00|4.00|-0.14|
> ||kNN|0.55±0.03|0.64±0.03|1.00|8.08|-0.16|
> |Diabetes|random|0.61±0.06|0.70±0.05|0.98|0.62|-0.36|
> ||kNN|0.46±0.02|0.75±0.00|1.00|0.44|-0.37|
>
> Our experimental results demonstrate that data synthesis using random sampling consistently underperforms our KNN-based approach, with an average decrease of 23% in performance on German dataset. On the diabetes dataset, our KNN-based approach also performs better on the LLE metric. This suggests that LLMs, when synthesizing data, still require a degree of similarity and context to effectively learn the relationships between data points. Random sampling, by introducing less relevant data, may hinder the LLM's ability to capture these relationships. Thanks for your suggestion again, this ablation study effectively demonstrates the efficacy of our method.
>
>
> **Suggestion 2: Learning the correlation between columns**
>
> Thank you for your suggestion. This may be a misunderstanding that our method can learn the correlation between columns. As we discuss in our related work of other LLM methods (lines 79-93) and our own method (lines 141-149), the input of each data point, including its feature permutation, allows LLMs to learn the correlation between columns. This has been experimentally validated in these LLM methods.
>
> To further clarify and address potential reader confusion, we will explicitly emphasize this point in both our related work section and our method description in the revised version.
>
> **Suggestion 3: Formal garantee of privacy**
>
> Thank you for your suggestion. To clarify the misunderstanding, our method is theoretically grounded. Existing literature [1,2] has preliminary evidence, both theoretical and experimental, suggesting that SFT-based methods retain far less memory of training data compared to continued pre-training. We will cite these in the revision to support our method.
>
> We also acknowledge that the rapid development of LLM attacks, which aim to exploit vulnerabilities in language models through methods like carefully crafted prompts, may potentially weaken the safety of our method. Therefore, as mentioned in Question 1, we will further investigate the integration of differential privacy and other techniques in our future work to enhance the security of our method.
>
> [1] Zhu, Zeyuan Allen, and Yuanzhi Li. "Physics of language models: Part 3.1, knowledge storage and extraction." arXiv preprint arXiv:2309.14316 (2023).
>
> [2] Jain, Samyak, et al. "How does fine-tuning affect your model? Mechanistic analysis on procedural tasks." R0-FoMo: Robustness of Few-shot and Zero-shot Learning in Large Foundation Models.

---

> > ### Comment · Reviewer_N5C1 · 2024-08-28
> >
> > Thank you for the detailed rebuttal. I believe that my concerns are well addressed, I'm glad to accept the paper.

---

> > > ### Author Rebuttal · Authors · 2024-08-28
> > >
> > > Thanks for your acknowledgment of our work and the improvements we've made.  Your feedback is really helpful for us to improve our work. We will revise our final version based on your feedback and the improvements we made during the rebuttal.

---

> ### Author Rebuttal · Authors · 2024-08-17
>
> To further explore **"Suggestion 2: Learning the correlation between columns"**, we also compute the **"CorrelationSimilarity"** metric [1] on the German and Diabetes datasets. This metric measures the correlation between a pair of numerical columns and assesses the similarity between the real and synthetic data. A score of 1 indicates that the pairwise correlations of the real and synthetic data are identical. For each dataset, we selected the two columns with the highest significance levels in relation to the target variable for calculation.
>
> | Data     | HARMONIC | SMOTE | TVAE | CTAB | TabDDPM | TABSYN | RTF | GReaT |
> |:----------:|:----------:|:-------:|:------:|:------:|:---------:|:--------:|:-----:|:-------:|
> | German   | 0.96     | 0.97  | 0.99 | 0.98 | 0.90    | 0.98   | 0.98| 0.98  |
> | Diabetes | 0.95     | 0.96  | 0.98 | 0.97 | 0.99    | 0.98   | 0.99| 0.88  |
>
> The experimental results demonstrate that our method can capture the relationships between columns, achieving comparable scores with other methods. Notably, on the Diabetes dataset, our method (0.95) significantly outperforms GReaT (0.88), another LLM-based approach.
>
> [1] https://docs.sdv.dev/sdmetrics/metrics/metrics-glossary/correlationsimilarity

---

> ### Author Rebuttal · Authors · 2024-08-24
>
> # **More Results**
>
> We conducted more comprehensive and systematic experiments for **"Suggestion 2"** and **"Question 1"**, incorporating metrics for evaluating inter-column correlations and the differentially private method PrivBayes. The results of these experiments are presented below.
>
> ### **Suggestion 2: Learning the correlation between columns**
>
> As you suggested, we have investigated the correlation between columns by adding a 'CorrelationSimilarity' metric. Now, we have computed this across all four datasets in our paper, and the results are presented below. The experimental results align with our previous response, demonstrating that our method is competitive with other approaches in effectively preserving inter-column correlations.
>
> | Dataset | HARMONIC | SMOTE | TVAE | CTABGAN | TabDDPM | TABSYN | RTF | GReaT |
> |---|---|---|---|---|---|---|---|---|
> | German | 0.96 | 0.97 | 0.99 | 0.98 | 0.90 | 0.98 | 0.98 | 0.98 |
> | Diabetes | 0.95 | 0.96 | 0.98 | 0.97 | 0.99 | 0.98 | 0.99 | 0.88 |
> | Adult | 0.90 | 0.99 | 0.98 | 0.97 | 0.99 | 0.97 | 0.99 | 0.94 |
> | Buddy | 0.93 | 0.98 | 0.97 | 0.99 | 1.00 | 1.00 | 0.98 | 1.00 |
>
>
> ### **Question 1: Discuss more related work (lack differentially private method)**
>
> While our method does not conflict with differential privacy approaches, we still supplemented the PrivBayes data synthesis method on the diabetes dataset to explore our method's effect. Although our method exhibits a lower DCR (privacy metric) compared to PrivBayes with differential privacy enabled, our approach consistently outperforms PrivBayes in terms of statistical distribution characteristics and the utility of the synthetic data and achieve near-identical results to MLE. This is because differential privacy prioritizes strong privacy guarantees, often at the expense of performance. At the ssame time, our method also achieves significantly improved privacy compared to other existing approaches without compromising effectiveness.
>
> |                | data_mismatch | wasserstein_dist | CorrelationSimilarity | MLE       | LLE       | NRS  | DCR  |
> |----------------|---------------|------------------|-------------------|-----------|-----------|------|------|
> | PrivBayes      | 0.10 ± 0.00    | 0.34 ± 0.01      | 0.88              | 0.48 ± 0.04| 0.20 ± 0.02| 1.00 | 0.82 |
> | HARMONIC       | **0.03 ± 0.05**    | **0.14 ± 0.01**      | **0.95**              | 0.46 ± 0.02| **0.75 ± 0.00**| **1.00** | 0.44 |

---

### Official Review · Reviewer_povU · 2024-07-09
**Method and benchmark for generating synthetic data under privacy considerations**

**Rating:** 6
**Confidence:** 5
**Correctness:** The main claims seem to be correct, b…
**Clarity:** The work is clearly written.

**Review:**

Strengths:

1. The study is well-structured and provides a comprehensive evaluation of the proposed framework; it includes detailed methodology, thorough experiments, and clear benchmarks in comparison with existing methods.
2. The integration of LLMs for tabular data synthesis and the focus on privacy has some novelty (e.g., specifically the development of metrics like LLM Efficacy and Data Leakage Test).
3. The work addresses a critical issue in data synthesis (i.e., balancing utility and privacy) which is highly relevant in many fields.
4. Thorough benchmarking against multiple methods on various datasets.

Weaknesses:

1. The method itself is not really novel.
2. The framework is primarily focused on classification tasks; what about regression?
3. The reliance on large pre-trained LLMs might not be practical after all, due to computational resource constraints.
4. More details (e.g., on configuration and implementation) could improve reproducibility.
5. The suitability of the framework data synthesis for numeric tasks and regression is not clear, which raises questions about the broader applicability.
6. Important works on tabular data generation have not been discussed or cited (e.g., "Language models are realistic tabular data generators" by Borisov et al., or "Large language model as attributed training data generator: A tale of diversity and bias" by Yu et al., or "Tabllm: Few-shot classification of tabular data with large language models" by Hegselmann et al.)

**Strengths:**

See strengths above.

**Additional Feedback:**

--

**Documentation:**

Could be further improved. See comments above.

**Ethics:**

--

**Limitations:**

A more detailed discussion on the ethical implications of using synthetic data, particularly in sensitive domains like healthcare and finance, would be beneficial. This could include potential misuse scenarios and how to mitigate them. The same holds for potential biases in synthetic data generation. How does HARMONIC ensure fairness? How does it avoid perpetuating existing biases in the training data?

**Opportunities For Improvement:**

Reproducibility: Need for detailed configuration and implementation details.
Incremental novelty: Builds heavily on existing techniques.
Resource requirements: High computational demands (i.e., reliance on large LLMs).
Scope: Focuses mainly on classification tasks, with limited emphasis on numeric tasks and regression.
Prior work: Important works on tabular data generation have not been cited (e.g., "Language models are realistic tabular data generators" by Borisov et al., or "Large language model as attributed training data generator: A tale of diversity and bias" by Yu et al., or "Tabllm: Few-shot classification of tabular data with large language models" by Hegselmann et al.)

**Relation To Prior Work:**

This is the weakest aspect of the work. See weaknesses above.

**Summary And Contributions:**

The article presents HARMONIC, a framework for generating synthetic tabular data using LLMs while addressing privacy concerns.
HARMONIC shows comparable performance to real data in most cases, with lower standard deviation and better stability.
HARMONIC demonstrates superior privacy protection compared to existing methods, particularly against data leakage.

---

> ### Author Rebuttal · Authors · 2024-08-16
>
> Thank you for your acknowledgement of our paper's contributions, experiments, and practical meanings. We also deeply appreciate your insightful suggestions, which will help us improve the quality of our work. We will address each of your points below and hope that these responses adequately address your concerns.
>
> **Question 1: Our method's novelty**
>
> Thank you for your question. We would like to further clarify the novelty of our method. As you have pointed out, our research addresses a critical issue in data synthesis, balancing utility and privacy, which is highly relevant across many fields. Existing LLM-based data synthesis methods, while achieving promising performance, often suffer from significant privacy risks due to their reliance on pre-training (line 37, lines 88-92).
>
> Therefore, our work uniquely constructs a framework to ensure LLMs achieve both high-quality tabular synthetic data generation and enhanced privacy. This framework encompasses leveraging the novel idea of kNN to LLM for data processing, first employing SFT to this field instead of traditional pretraining method for privacy-preserving data generation, and the establishment of new evaluation metrics for both effectiveness and privacy. We believe these contributions demonstrate the novelty of our research.
>
> **Question 2: Evaluation on the regression tasks**
>
> Thank you for your feedback. We have conducted these additional experiments, the results of which are presented in the following Table. Due to computational and time constraints, we will update the discussion section with these findings throughout the disccusion process. Following the standard procedure for this track, we will also incorporate these results in the revised version.
>
> Currently, we have expanded a regression dataset, Abalone [1] for our experiments. We will further enrich our analysis by incorporating 1-2 additional regression datasets during the discusstion period. In the regression task, the MLE and LLE metrics are evaluated using R2 score rather than F1 score. All other aspects of the experimental framework, including procedures and configurations, remain identical to those outlined in the main text.
>
> | Data   | Metric | Original    | HARMONIC      | SMOTE         | TVAE          | CTAB | TabDDPM       | TABSYN        | RTF            | GReaT          |
> |--------|--------|-------------|---------------|---------------|---------------|------|---------------|---------------|----------------|----------------|
> | Abalone| MLE    | 0.42±0.00   | 0.24±0.02     | 0.40±0.01     | 0.22±0.03     | -    | 0.35±0.02     | 0.33±0.01     | 0.33±0.02      | 0.09±0.04      |
> |        | LLE    | 0.39±0.00   | 0.22±0.03     | 0.25±0.03     | 0.00±0.17     | -    | 0.12±0.18     | 0.21±0.10     | -32.67±46.36   | 0.27±0.03      |
> |        | NRS    | -           | 1.00          | 0.88          | 1.00          | -    | 1.00          | 1.00          | 1.00           | 1.00           |
> |        | DCR    | -           | 0.18          | 0.05          | 0.18          | -    | 0.14          | 0.13          | 0.11           | 0.15           |
> |        | DLT    | -           | -0.38         | -             | -             | -    | -             | -             | -68.99            | -0.71          |
>
>
>
> Our experimental results demonstrate that our method not only maintains strong privacy protection but also exhibits better stability in regression tasks compared to other LLM-based tabular synthetic methods. We achieve a higher DCR (0.18 vs. 0.15 for GReaT and 0.11 for RTF) and a higher DLT (-0.38 vs. -0.71 for GReaT and -68.99 for RTF), signifying our ability to maintain privacy even for regression tasks. Similarly, our method displays consistent performance in terms of data quality, measured by LLE and MLE metrics, while RTF exhibits significantly worse performance in LLE (-32.67), and GReaT experiences a considerable decrease in MLE (0.09).
>
> We also would like to clarify why we did not initially consider testing on regression data in our first version. When we first considered the setting, relevant references [2,3] indicated that LLMs are not adept at numerical computations. Therefore, we initially believed that this was not a scenario where LLMs could leverage their strengths (features and in-context understanding) to generate synthetic tabular data for regression tasks [4].
>
> [1] https://www.openml.org/search?type=data&sort=runs&id=183&status=active
>
> [2] Shen, Ruoqi, et al. "Positional description matters for transformers arithmetic." arXiv preprint arXiv:2311.14737 (2023).
>
> [3] Dziri, Nouha, et al. "Faith and fate: Limits of transformers on compositionality." Advances in Neural Information Processing Systems 36 (2024).
>
> [4] Borisov, Vadim, et al. "Language models are realistic tabular data generators." arXiv preprint arXiv:2210.06280 (2022).

---

> ### Author Rebuttal · Authors · 2024-08-16
>
> **Question 3: Computational resource constraints**
>
> Thank you for your feedback. We acknowledge that training and deploying LLMs currently present certain bottlenecks for individuals, but we have taken steps to mitigate these challenges. First, we utilizes LoRA to minimize training load and hardware requirements (line 367), requiring only a single A100 40G GPU (with the potential for further resource reduction). Moreover, current applications of synthetic data primarily focus on companies and government (line 24), which have the computational resources necessary for our methods.
>
> Take a step forward, we believe our method will become more applicable to a wider range of contexts as hardware performance improves and cloud computing advances. Just like the BERT model, initially difficult to run on many personal computers, is now widely accessible.
>
> **Question 4: More details**
>
> Thank you for your feedback. We will provide additional details in Appendix A and B. We would appreciate your further feedback on the specifics of our experimental setup, particularly regarding any aspects you feel require elaboration. In addition, our github link is now public and available at https://github.com/Wendy619/HARMONIC.
>
> **Question 5: Prior work**
>
> Thank you for your feedback. We would like to clarify your misunderstanding.
>
> We have reviewed some of the related works you mentioned in our paper and conducted experiments. The first reference by Borisov et al. that you recommended, "Language models are realistic tabular data generators" (GReaT), has been reviewed in our paper (Lines 34 and 82) and we used it as a baseline in our experiments (Line 249).
>
> We will further discuss other relevant works in our related work section in the final revision. The two additional references that you recommended, one uses prompt engineering for synthetic data generation (it does not mainly focus on the tabular data). The other, TabLLM, explores the application of LLMs for tabular data classification. These methods utilize the in-context capability of LLMs, which shares similarities with our approach to leveraging the advantages of LLMs. However, our method specifically focuses on the generation of synthetic tabular data.
>
> **Suggestion 1: Discussion for ethical and potential biases**
>
> Thank you for your suggestion. In the revised manuscript, we will expand upon the limitations section to discuss these in detail.
>
> The ethical and potential biases of synthetic data is a critical and complex area in the field. On the one hand, the demand for synthetic data arises from how to use the data with the concerns about data privacy and the potential for data leaks. Thus, there is a strong desire for synthetic data to accurately represent the characteristics of the original data. However, on the other hand, achieving a faithful representation can introduce potential biases present in the original data, which may be difficult to identify solely through synthetic data. Attempting to mitigate this bias can lead to a decline in synthetic data quality, rendering it unsuitable for downstream tasks. Striking a balance between realism and ethical considerations is crucial, and we recognize that addressing this challenge is an ongoing area of research.

---

> ### Author Rebuttal · Authors · 2024-08-17
>
> Thank you again for your feedback. Based on your **"Suggestion 1: Discussion for ethical and potential biases"**, we plan to revise the Limitations section of the paper in our manuscript as follows. Please let us know if you have any concerns, and we are always open to further discussion on how to further improve this paper.
>
> (1) Computational Resources: Our method requires longer processing times for larger LLMs compared to other approaches.
>
> (2) Focus on Classification: Due to LLMs' limited sensitivity to numerical data and their suitability for classification tasks rather than regression tasks, our current work focuses on tabular data for classification tasks.
>
> **(3) Ethical and Bias Concerns: While the generation and use of synthetic data offer valuable benefits, they also raise ethical concerns, particularly when dealing with sensitive information. Moreover, biases present in the original data can be inherited and even amplified in the synthetic data, potentially affecting downstream tasks. However, these challenges often stem from the complexity of the data source domain, necessitating further collaboration between synthetic data creators and domain experts.**

---

> ### Author Rebuttal · Authors · 2024-08-24
>
> Thank you for reviewing our paper. We hope we have comprehensively addressed all your concerns based on your feedback. Please do not hesitate to contact us if you have any remaining questions, issues, or points requiring further discussion.

---

> > ### Comment · Reviewer_povU · 2024-08-26
> > **Thank you for the improvements.**
> >
> > Thank you for the detailed improvements based on my feedback. I still think the work lacks novelty. Nevertheless, I increased my final score to reflect the improvements.

---

> > ### Author Rebuttal · Authors · 2024-08-27
> >
> > Thank you for acknowledging our work and the improvements we've made, and for giving us a positive score. To clarify your concern regarding the "our work lacks novelty", our research makes three key contributions that, to the best of our knowledge, are novel:
> >
> > **1. Propose a novel framework for LLM-based tabular data synthesis:** Our proposed framework is the first to leverage Supervised Fine-Tuning (SFT) with kNN based instruction tuning datasets for generating tabluar synthetic data using LLMs. This innovative approach allows us to generate high-quality synthetic tabular data while enhancing privacy, addressing a critical issue in data synthesis.
> >
> > **2. Introduce Novel Evaluation Metrics:** We introduce two novel metrics to evaluate both the effectiveness and privacy of synthetic data generated by our method. These metrics are the first to provide a more robust and nuanced assessment of generated data than existing approaches. Additionally, we have incorporated reviewer feedback by adding more metrics that provide a more intuitive assessment of synthetic data quality.
> >
> > **3. More Comprehensive Evaluation of Existing Tabular Data Synthesis Methods:** We conducted a comprehensive evaluation of existing tabular data synthesis methods, including simple interpolation, GAN-based, diffusion-based, transformer-based, and LLM-based approaches. Our evaluation also considered three key aspects: direct statistical comparison with the original data (suggested by reviewer YZek), downstream task performance, and privacy of the generated data. Our results demonstrate that previous methods often require trade-offs between data quality and privacy, while our method achieves a better balance between these two crucial aspects.
> >
> > These contributions significantly advance the field of data synthesis and align with the concerns of this Datasets and Benchmarks Track. We appreciate your feedback and are committed to further refining our work to ensure these novelties are clearly presented.

---

### Official Review · Reviewer_YZek · 2024-07-20
**LLMs for synthesizing private tabular data**

**Rating:** 7
**Confidence:** 5

**Review:**

This work presents a solid novel approach based on fine-tuning that achieves state-of-the-art or close-to-the-state-of-the-art performance regarding utility and improves the state-of-the-art regarding privacy. The approach is to the best of my knowledge novel and original. Increasing privacy without significantly compromising utility is an important endeavour.

Detailed comments:
Line 44: "generating" instead of "generate".
Line 54: reformulate sentence - it does not make sense like this.
Table 2: You forgot bold and underline for DI-PPL.
Line 259: space missing after "NRS".
Line 320: reformulate the sentence - it does not make sense like this.

**Strengths:**

The proposed data synthesis approach achieves good utility and excellent privacy. The benchmark includes a sufficiently broad comparison with other methods.

**Additional Feedback:**

None.

**Clarity:**

The paper is mostly well-written. The language needs some proofreading, though. I pointed out some instances in the detailed comments above.

**Correctness:**

The code is not available (I get a 404 for the github link). Thus, neither dataset nor code are available for assessment. I did not detect major technical issues in the manuscript itself.

**Documentation:**

No, the datasets was not available. Nor did I find a hosting, licensing, or maintenance plan.

**Ethics:**

No.

**Limitations:**

Partially. The manuscript should acknowledge that its selection of data synthesis and evaluation methods only covers a minor fraction of the full spectrum.

**Opportunities For Improvement:**

The benchmark misses important approaches such as PrivBayes (private data synthesis using Bayesian Networks). These have also been shown to reach good utility levels with excellent privacy levels. Given the focus of this manuscript, this would have been an obvious candidate to include.

Another opportunity for improvement would have been a broader selection of metrics. These are readily available in tools such as syntheval or synthcity:
https://github.com/schneiderkamplab/syntheval
https://github.com/vanderschaarlab/synthcity

**Relation To Prior Work:**

Yes.

**Summary And Contributions:**

The manuscript describes a fine-tuning strategy for LLMs to generate tabular data with good privacy characteristics. The manuscript contains a benchmark against some non-LLM- and some LLM-based data synthesis methods.

---

> ### Author Rebuttal · Authors · 2024-08-16
>
> Thank you for your acknowledgement of our work and your valuable suggestions. We are incorporating your feedback into our paper and would like to explain in detail how we will address your suggestions point by point.
>
> **Question 1: Reformulate the sentence & typoes**
>
> Thank you for your thorough review. We will address all the typos and writing issues you have raised in the final revision.
>
> **Question 2: Github link invalid & datasets not available**
>
> Thank you for pointing out this oversight. We had previously set the project to private, but it is now public and available at https://github.com/Wendy619/HARMONIC.
>
> **Suggestion 1: Compare to PrivBayes**
>
> Thank you for your suggestion. We will incorporate two key improvements based on your suggestion in the revision.
>
> Firstly, we will include PrivBayes, along with other differentially private privacy-preserving methods recommended by other reviewers, in our related work section for a comprehensive review and comparison. While these differentially private methods offer a path towards privacy preservation on top of existing data synthesis methods, we believe our method, which directly modifies the data synthesis method itself to enhance privacy, does not conflict with them. In fact, our method can be further integrated with differentially private techniques, and we will discuss this potential in the future work section.
>
> Secondly, we will conduct comparative experiments as you suggested. However, due to computer source limitations, we will provide updated results and analysis throughout this discussion round and in the revision.
>
> **Suggestion 2: More metrics**
>
> Thank you for your suggestion. We will also incorporate a review of this topic in the related work section, and showcase relevant results in the revision and in this discussion round through comments.
>
> Currently, we have implemented two metrics, "data_mismatch" and "wasserstein_dist", on the German and Diabetes datasets. "Data_mismatch" measures the average number of columns with datatype (object, real, int) mismatch between the real and synthetic data. A value of 0 indicates no datatype mismatch, while 1 represents complete data type mismatch between the datasets. "Wasserstein_dist" is a well-known measure of the distance between two probability distributions, where 0 indicates identical distributions. More results of these metrics will be added in the revision and comments during the discussion period. Thanks for your suggestion again, this will add new dimensions and perspectives to our existing evaluation framework, enhancing its comprehensiveness.
>
> | Data      | Metric  | HARMONIC     | SMOTE        | TVAE         | CTAB         | TabDDPM      | TABSYN       | RTF          | GReaT        |
> |:--:|:--:|:--:|:--:|:--:|:--:|:--:|:--:|:--:|:--:|
> | German    | data_mismatch       | 0.00 ± 0.00  | 0.14 ± 0.00  | 0.14 ± 0.00  | 0.14 ± 0.00  | 0.14 ± 0.00  | 0.27 ± 0.00  | 0.00 ± 0.00  | 0.14 ± 0.00  |
> |           | wasserstein_dist     | 0.87 ± 0.07  | 0.85 ± 0.03  | 0.70 ± 0.04  | 0.77 ± 0.02  | 0.73 ± 0.02  | 0.94 ± 0.09  | 0.67 ± 0.01  | 0.93 ± 0.22  |
> | Diabetes  | data_mismatch       | 0.03 ± 0.05  | 0.07 ± 0.05  | 0.10 ± 0.00  | 0.10 ± 0.00  | 0.10 ± 0.00  | 0.10 ± 0.00  | 0.10 ± 0.00  | 0.07 ± 0.05  |
> |           | wasserstein_dist     | 0.14 ± 0.01  | 0.07 ± 0.00  | 0.07 ± 0.00  | 0.26 ± 0.01  | 0.08 ± 0.00  | 0.09 ± 0.01  | 0.09 ± 0.02  | 0.13 ± 0.00  |
>
> The experimental results demonstrate that our method achieves state-of-the-art performance in terms of generated data sanity across all synthetic data generation methods, while remaining comparable to other LLM-based approaches in terms of its proximity to the original distribution. Our method achieves a "data_mismatch" score of 0 for the German dataset and 0.03 for the Diabetes dataset, indicating that our generated data exhibits virtually no formatting errors, surpassing all other methods. On the "wasserstein_dist" metric, we perform comparably to LLM-based methods like GReaT (0.87 vs. 0.93 for German, 0.14 vs. 0.13 for Diabetes), demonstrating our ability to effectively preserve the original data distribution. These overall findings further validate the effectiveness of our method.

---

> ### Author Rebuttal · Authors · 2024-08-24
>
> # **More Results**
>
> We have conducted more experiments for **"Suggestion 2"** and **"Suggestion 1"**, incorporating additional experimental metrics and tabular data synthesis methods. The results of these experiments are presented below.
>
> ### **Suggestion 2: More metrics**
>
> In addition to the two metrics we previously introduced, "data_mismatch" and "wasserstein_dist," we have also incorporated the "CorrelationSimilarity" [1] metric. This metric measures the correlation between a pair of numerical columns and assesses the similarity between the real and synthetic data. Unlike our previous metrics like MLE and LLE, which focus on the downstream utility of tabular synthetic data, these three metrics are more concerned with the statistical properties of the synthetic data itself.
>
> Now, we show the results of all three metrics across the four datasets in our main experiment. The experimental results show that our method achieves comparable or even superior performance to existing state-of-the-art models across all three metrics.
>
> | Data | Metric | HARMONIC | SMOTE | TVAE | CTABGAN | TabDDPM | TABSYN | RTF | GReaT |
> |---|---|---|---|---|---|---|---|---|---|
> | German | data_mismatch | 0.00±0.00 | 0.14±0.00 | 0.14±0.00 | 0.14±0.00 | 0.14±0.00 | 0.27±0.00 | 0.00±0.00 | 0.14±0.00 |
> | | wasserstein_dist | 0.87±0.07 | 0.85±0.03 | 0.70±0.04 | 0.77±0.02 | 0.73±0.02 | 0.94±0.09 | 0.67±0.01 | 0.93±0.22 |
> | | CorrelationSimilarity | 0.96 | 0.97 | 0.99 | 0.98 | 0.90 | 0.98 | 0.98 | 0.98 |
> | Diabetes | data_mismatch | 0.03±0.05 | 0.07±0.05 | 0.10±0.00 | 0.10±0.00 | 0.07±0.05 | 0.10±0.00 | 0.10±0.00 | 0.07±0.05 |
> | | wasserstein_dist | 0.14±0.01 | 0.07±0.00 | 0.07±0.00 | 0.26±0.01 | 0.08±0.00 | 0.09±0.01 | 0.09±0.02 | 0.13±0.00 |
> | | CorrelationSimilarity | 0.95 | 0.96 | 0.98 | 0.97 | 0.99 | 0.98 | 0.99 | 0.88 |
> | Adult | data_mismatch | 0.21±0.15 | 0.00±0.00 | 0.00±0.00 | 0.00±0.00 | 0.00±0.00 | 0.00±0.00 | 0.00±0.00 | 0.06±0.00 |
> | | wasserstein_dist | 0.48±0.15 | 0.49±0.01 | 0.31±0.04 | 0.07±0.01 | 0.06±0.01 | 0.07±0.01 | 0.03±0.00 | 3.83±0.09 |
> | | CorrelationSimilarity | 0.90 | 0.99 | 0.98 | 0.97 | 0.99 | 0.97 | 0.99 | 0.94 |
> | Buddy | data_mismatch | 0.00±0.00 | 0.00±0.00 | 0.00±0.00 | 0.00±0.00 | 0.00±0.00 | 0.00±0.00 | 0.00±0.00 | 0.03±0.04 |
> | | wasserstein_dist | 0.48±0.16 | 0.23±0.02 | 0.10±0.01 | 0.05±0.00 | 0.06±0.02 | 0.04±0.00 | 0.04±0.00 | 2292.47±1014.22 |
> | | CorrelationSimilarity | 0.93 | 0.98 | 0.97 | 0.99 | 1.00 | 1.00 | 0.98 | 1.00 |
>
> [1] https://docs.sdv.dev/sdmetrics/metrics/metrics-glossary/correlationsimilarity
>
> ### **Suggestion 1: Compare to PrivBayes**
>
> While our method does not conflict with differential privacy approaches, we still supplemented the PrivBayes data synthesis method on the diabetes dataset to explore our method's effect. The results are shown below. (Here, all metrics in the experimental results are better with higher values, except for data_mismatch and wasserstein_dist.)
>
> |                | data_mismatch | wasserstein_dist | CorrelationSimilarity | MLE       | LLE       | NRS  | DCR  |
> |----------------|---------------|------------------|-------------------|-----------|-----------|------|------|
> | PrivBayes      | 0.10 ± 0.00    | 0.34 ± 0.01      | 0.88              | 0.48 ± 0.04| 0.20 ± 0.02| 1.00 | 0.82 |
> | HARMONIC       | **0.03 ± 0.05**    | **0.14 ± 0.01**      | **0.95**              | 0.46 ± 0.02| **0.75 ± 0.00**| **1.00** | 0.44 |
>
> The results demonstrate that our method, HARMONIC, although exhibits a lower DCR (privacy metric) compared to PrivBayes with differential privacy enabled, it consistently outperforms PrivBayes in terms of statistical distribution characteristics and the utility of the synthetic data and achieve near-identical results to MLE. This is because differential privacy prioritizes strong privacy guarantees, often at the expense of performance. At the ssame time, our method also achieves significantly improved privacy compared to other existing approaches without compromising effectiveness.
>
> Furthermore, it is important to note that the recommended implementation of the PrivBayes project [1] currently faces significant computational time issues, requiring extensive time to generate a single dataset.
>
> [1] https://github.com/vanderschaarlab/synthcity

---

### Author Rebuttal · Authors · 2024-08-30

We sincerely thank all reviewers for their time and detailed feedback. We are grateful for the reviewers' acknowledgement of our paper and its contributions. The reviewers all agree that our exploration of privacy-enhancing techniques of synthetic data method, without sacrificing utility, is crucial and aligns perfectly with the Datasets and Benchmarks Track's focus. They also all agree that our study is well-structured and provides a comprehensive evaluation of the proposed framework. Additionally, most reviewers (Reviewer YZek, N5C1) commend our study as a solid and novel approach for synthetic tabular data.

We also appreciate the opportunity to reemphasize our contributions:

1. We propose a novel framework for LLM-based synthetic tabular data that first uses supervised fine-tuning (SFT) with kNN constrcuted instruction tuning dataset, enabling the generation of high-quality synthetic data while enhancing privacy.
2. We introduce two novel metrics to evaluate the effectiveness and privacy of synthetic data, providing a more robust and nuanced assessment.
3. We conduct a comprehensive evaluation of existing tabular data synthesis methods, demonstrating that our framework achieves a better balance between data quality and privacy compared to previous approaches.

To further address the reviewers' concerns, we have provided detailed responses in the rebuttal, which can be summarized into the following four key points (the results are shown in the PDF):

1. **More Evaluation Metrics:** We further added three additional evaluation metrics to our main experiments across all four datasets to directly assess the quality of the synthetic data (rather than the downstream evaluation in our original paper). The results show that our approach achieves comparable or even better performance than existing state-of-the-art models on all three metrics.
2. **More Ablation Study:** We further conducted an ablation study of our framework. A control group was established by randomly selecting five data points for each instance without using the KNN method. The experimental results further demonstrate the effectiveness of our KNN component in our framework, which enhances the framework's ability to capture information between the data.
3. **Regression Experiments:** Although relevant references [1, 2] have shown that LLMs are not suitble for handling numerical data in regression tasks, we also added regression experiments as requested. The results demonstrate that our method also performs well, even when applied to regression tasks. This further indicates that our framework may improve LLMs to generate synthetic data for regression tasks.
4. **Discussion of Differential Privacy:** We will discuss differential privacy in the related work section. We reiterate that our approach is inherently privacy-preserving (Enhancing privacy preservation through synthetic tabular data method itself) and can be further enhanced by integrating with differential privacy techniques. Therefore, it is important to note that differential privacy based methods may not be viewed as competitors to our method. To further illustrate this point, we have included an additional experiment using the PrivBayes, a differential privacy method. The results consistently show that while differential privacy offers strong privacy guarantees, it can significantly hinder the data generation capabilities of the original method, ultimately leading to inferior performance compared to our framework, which strikes a balance between effectiveness and privacy.

In summary, our responses address all reviewer concerns, and we are committed to incorporating these improvements into our final revisions to further strengthen our work.

[1] Shen, Ruoqi, et al. "Positional description matters for transformers arithmetic." arXiv preprint arXiv:2311.14737 (2023).

[2] Dziri, Nouha, et al. "Faith and fate: Limits of transformers on compositionality." Advances in Neural Information Processing Systems 36 (2024).

---

### Decision · Program_Chairs · 2024-09-26

**Decision:**

Accept (Poster)

**Comment:**

The paper is focusing on using LLM for tabular data generation. All reviewers agree that the proposed data synthesis approach achieves good utility and excellent privacy. The benchmark includes a sufficiently broad comparison with other methods. So all reviewers agree to accept the paper. Thus, I vote for an acceptance.